# Dose Effect of Polyethylene Microplastics Derived from Commercial Resins on Soil Properties, Bacterial Communities, and Enzymatic Activity

**DOI:** 10.3390/microorganisms12091790

**Published:** 2024-08-29

**Authors:** Lesbia Gicel Cruz, Fo-Ting Shen, Chiou-Pin Chen, Wen-Ching Chen

**Affiliations:** 1International Master Program in Agriculture, National Chung Hsing University, Taichung 40227, Taiwan; giz_cruz@yahoo.com; 2Department of Soil and Environmental Sciences, National Chung Hsing University, Taichung 40227, Taiwan; ftshen@dragon.nchu.edu.tw; 3Innovation and Development Center of Sustainable Agriculture (IDCSA), National Chung Hsing University, Taichung 40227, Taiwan; 4The Experimental Forest, College of Bio-Resources and Agriculture, National Taiwan University, Nantou County 557004, Taiwan; chioupinchen@ntu.edu.tw; 5International Bachelor Program in Agribusiness, College of Agriculture and Natural Resources, National Chung Hsing University, Taichung 40227, Taiwan

**Keywords:** soil contaminant, soil pollutant, plastics, plastic mulch, dose–response effect, soil environment, soil microbe, soil microbial ecology, soil microbial diversity, soil enzyme

## Abstract

Soils are the largest reservoir of microplastics (MPs) on earth. Since MPs can remain in soils for a very long time, their effects are magnified. In this study, different concentrations of polyethylene (PE) MPs derived from commercial resins (0%, 1%, 7%, and 14%, represented as MP_0, MP_1, MP_7, and MP_14) were added to soils to assess the changes in the soils’ chemical properties, enzyme activities, and bacterial communities during a 70-day incubation period. The results show that PE MP treatments with low concentrations differed from other treatments in terms of exchangeable Ca and Mg, whereas at high concentrations, the pH and availability of phosphate ions differed. Fluorescein diacetate (FDA), acid phosphatase (ACP), and N-acetyl-β-d-glucosaminidase (NAG) enzyme activities exhibited a dose-related trend with the addition of the PE MPs; however, the average FDA and ACP activities were significantly affected only by MP_14. Changes in the microbial communities were observed at both the phylum and family levels with all PE MP treatments. It was revealed that even a low dosage of PE MPs in soils can affect the functional microbes, and a greater impact is observed on those that can survive in polluted environments with limited resources.

## 1. Introduction

Microplastics (MPs), which range from 5 mm to 1 µm in size, have been identified as a common soil contaminant of great concern; therefore, their impacts on soil environments require thorough study [1,2,3]. MPs such as polyethylene (PE) and polypropylene (PP) are introduced into soils by anthropogenic activities such as sewage sludge fertilization, plastic mulches, irrigation, street runoff, environmental deposition, and the improper disposal of plastic debris [4,5]. Among soil environments, farmland soil is of high concern due to its relevance to plant production and human health.

Nizzetto et al. [6] estimated that the annual input of MPs by farmlands is 63–430 thousand tons in Europe and 44–300 thousand tons in North America. Zhang and Liu [7] reported that approximately 0.008–0.068 g kg^−1^ of MPs were found in an agricultural field in Northwest China. Huang et al. [8] also reported that approximately 1075 ± 346 pieces kg^−1^ soil of PE MPs had accumulated in agricultural soils owing to the extensive use of mulches over a 24-year period. Previous studies have indicated that MPs may impact soil chemical properties such as pH and physical properties such as soil porosity, structure, and water-holding capacity, thus inducing microbiome ecological changes [1,9,10,11,12]. However, these impacts varied from significant to nonsignificant [13]. Since MPs remain in the soil environment once they are introduced without being easily broken down, their effects on the soil environment can be magnified over time.

In addition to the impact of pure MPs, the weathering of MPs also releases additives which can potentially alter the chemical properties of soil and be toxic to organisms residing in soil [14,15]. MP additives are a factor that must be considered in soil environments. Many studies have focused on pure MPs without additives in soil environments. In this study, we used PE MPs derived from commercial resins to evaluate the possible influence of MPs and their additives. The aims of the present study were to identify soil chemical properties, soil enzyme activity, and soil bacterial community changes upon the addition of MPs to the soil of an organic farm that was free from MP contamination. In the meantime, different doses of MPs ranging from the level of industrial contamination to the level of accumulation in farmland were applied. Relevant studies which also focused on commercial PE MP fragments were compared and discussed to obtain the possible impact of the addition of PE MPs to soils.

## 2. Materials and Methods

### 2.1. Soil Collection and Preparation

Soil was collected from an organic rice farm in Wufong District, Taichung City, Taiwan (longitude: 24.0784, latitude: 120.7079). This area did not have a previous history of plastic mulch or sewage treatment use. The collected soil was stored and transported in aluminum containers. It was dried and sieved using a 2 mm metal sieve. The debris, rocks, and gravel were removed manually. The density separation method by flotation with concentrated NaCl and microscope visualization [16] were used to confirm that the soils were not contaminated by MPs before experiments. The soil was stored at 4 °C until one week before use.

### 2.2. Soil Incubation and PE MP Treatments

Linear low-density polyethylene plastic resin pellets (density: 0.922 g cm^−3^) were purchased from Formosa Plastics Corporation and were mechanically ground to create MPs. This resin can be used to produce plastic mulch for agricultural usage. The MP fragments were separated using a vibratory sieve shaker and sieves (AS 200 digit cA) into different size ranges: 500–355 µm, 355–250 µm, and 250–180 µm. MPs with sizes below 500 µm were chosen as these sizes are commonly observed in soil [7]. The MPs of different sizes were mixed in 1:1:1 ratio to obtain an equal representation of sizes.

The experiment involved four treatments with three replicates: MP_0 (control), MP_1, MP_7, and MP_14, in which the numbers indicate the weight percentage of PE MPs in the soil. These concentrations are chosen because of their environmental relevance: some agricultural soils are reported to contain 1% MPs [7], an industrial contaminant area was reported to contain approximately 7% MPs [17], and contamination levels have been the predicted to increase to 14% in the future due to continuous contamination [8]. The soil was pre-incubated for 7 days to recover its microbial ecology. Then, PE MP fragments were weighed and added to 700 g of soil in glass beakers to achieve the desired treatment concentrations. The ground MPs were washed with sufficient disinfected water, filtered, and dried overnight in an oven at 60 °C before being placed under a UV hood for 96 h to avoid microbial contamination. The experiment was conducted over 70 days at a constant temperature of 30 °C and soil moisture at 60% water-holding capacity (WHC). Water loss due to evaporation was measured and recovered by adding an equivalent weight of disinfected water every 2 days.

Rapid tests of pH and electric conductivity (EC) for PE MP leachates were also performed (Appendix A). Differential concentrations of MPs (0%, 1%, 7%, and 14%), which were either washed with disinfected H_2_O or were not washed, were mixed in the soil or in pure water before the analysis.

### 2.3. Analysis of Chemical Properties of Soil

The chemical properties of bulk soil were analyzed on day 0 (initial values in Table 1 and Table 2). The properties were analyzed following standard protocols from Sparks et al. [18]. The EC and pH were determined at soil:water ratio of 1:2 using an EC meter (SC 2300 Suntex Conductivity Meter, Suntex, Taiwan) and a pH meter (F20 Mettler Toledo Five Easy pH meter, Zürich, Switzerland). The soil organic matter and WHC were determined from the weight loss by ignition and water retention in soils at field capacity [19], respectively. The available N (AN) was determined by extraction using 2 M KCl and measured using a Kjeldahl Distillation apparatus. The available P (AP) was determined using Bray-1 extractant and measured using a spectrophotometer (Thermo Scientific Genesys 30 visible spectrophotometer, Waltham, MA, USA) at a wavelength of 650 nm. The total N (TN) and C/N ratio were analyzed using an elemental analyzer (Vario EL cube, Langenselbold, Germany). The cation exchange capacity (CEC) and exchangeable cations were determined by following the method from Carter and Gregorich [20]. Specifically, 1 M ammonium acetate was used to wash soil, and the extracts were saved for analyzing the exchangeable cations (Ca, Mg, K, and Na) using inductively coupled plasma atomic emission spectroscopy (ICP-AES; PerkinElmer Avio200, Waltham, MA, USA); further, the CEC was determined by the sum of exchangeable cations.

### 2.4. Enzyme Activities and Community-Level Physiological Profiling (CLPP) Analysis

The activities of the enzymes fluorescein diacetate (FDA), acid phosphatase (ACP), and N-acetyl-β-d-glucosaminidase (NAG) and CLPP were analyzed using spectrophotometers (Thermo Scientific Genesys 30, Waltham, MA, USA and Biochrom Asys UVM 340, Holliston, MA, USA) on days 1, 7, 14, 28, 42, and 70. FDA was characterized by following the protocol from Green et al. [21]. One g of soil was incubated for 3 h at 37 °C with 50 mL of 60 mM sodium phosphate buffer (pH 7.6) and 0.50 mL of 4.9 mM FDA lipase substrate solution. The reaction was stopped using 2 mL of acetone, and the fluorescein filtrate was measured at a wavelength of 490 nm. ACP was characterized by following the method reported by Weaver et al. [22]. One g of soil was incubated with 4 mL of modified universal buffer (pH 6.5) and 1 mL of substrate solution (p-nitrophenyl phosphatase) for 1 h at 37 °C. The reaction was stopped by adding 1 mL of 0.5 M CaCl_2_ and 4 mL of 0.5 M NaOH, and the filtrate (p-nitrophenol) was measured at a wavelength of 420 nm. NAG was characterized following the protocol by Parham and Deng [23]. For this purpose, 1 g of soil was incubated with 3 mL of acetate buffer and 1 mL of 10 mM ρ-nitrophenyl-NAG substrate for 1 h at 37 °C. The reaction was stopped by adding 1 mL of 0.5 M CaCl_2_ and 4 mL of 0.5 M NaOH, and the filtrate (p-nitrophenol) was measured at a wavelength of 420 nm.

CLPP was performed using Biolog^®^ Ecoplate^TM^ (Biolog Inc., Hayward, CA, USA) (Insam, 1997) [24] following the method reported by Sofo and Ricciuti [25] and Weber and Legge [26] with modifications. For this purpose, 1 g of soil in 9 mL of phosphate-buffered saline solution was serially diluted (10^−3^), and then, 160 µL of was inoculated in 30 carbon sources in 96 wells of Biolog^®^ Ecoplate^TM^ using a multichannel pipette. The absorbance of the plates was measured at 48 h using wavelengths of 590 and 750 nm for formazan color development on a scanning microplate reader. The plates were analyzed for average well color development (AWCD), diversity, richness, and evenness of substrates [25].

### 2.5. Colony-Forming Units (CFUs)

Ten mL of soil extract (1 g of soil in 9 mL of disinfected H_2_O) was incubated in a 90-mL nutrient broth in flasks containing 0%, 1%, 7%, and 14% (*w*/*w*) MPs which were 180–250 µm. The flasks were stoppered, placed in an incubator, and shaken at 180 rpm for 7 days at 30 °C. The culture solution was spread on nutrient-rich and nutrient-poor agar plates [27] using serial dilution technique to count the CFUs per milliliter of soil solution.

### 2.6. Next-Generation Sequencing (NGS) Metagenomics

The soil DNA was extracted using Fast DNA SPIN Kit (MP Biomedicals, Santa Ana, CA, USA) before DNA sequencing (Genomics BioSci&Tech Ltd., Taichung, Taiwan). The sequencing process is described briefly as follows. PCR amplicons were generated by amplifying the V3-V4 region of 16S rDNA using primer set 341F-805R with KAPA High-Fidelity PCR kit (KAPA BIOSYSTEMS, Cape Town, South Africa) and purified using QIAquick Gel Extraction Kit (QIAGEN, Germantown, MI, USA). Sequence libraries were generated using Truseq nano DNA Library Prep Kit (Illumina, San Diego, CA, USA) followed by sequencing on an Illumina Miseq platform producing 300 bp paired-end reads. Primer sequences were trimmed using the Cutadapt program and merged using FLASH software (v1.2.11). Trimmed and merged sequences were reanalyzed using FastQC (v0.11.5) and MultiQC (v0.9). Mothur (v1.39.5) was used for picking operational taxonomic units (OTUs) with 97% identity. An OTU table was produced by identifying chimera sequences using UCHIME (v4.2) software. Sequences in the same OTU were annotated using the SILVA database. The relative abundance expressed by OTU was used to determine the dominant bacterial group at the phylum level (>1%) and family level (>1.5%).

### 2.7. Statistical Analysis

Analysis of variance using a general linear model (multivariate) was performed at a significance level of *p* < 0.05 by the least significant difference post-hoc test (IBM SPSS Statistic version 25) for the effect of the treatments on soil properties, enzyme activities, biodiversity, NGS data, and CLPP. Principal component analysis (PCA) for soil properties was performed using XLSTAT 2019 software (Addinsoft Inc., New York, NY, USA).

## 3. Results

### 3.1. Influence of PE MPs on Chemical Properties of Soil

The bulk soil was classified as silt loam with 24% clay, 22% sand, and 54% silt. The soil was slightly acidic (with a pH of 6.73), with a 2.58 ± 0.16% organic matter content and an approximately 65% WHC. The other chemical properties of the bulk soil prior to incubation are listed in Table 1 and Table 2 as initial values.

Table 1 shows the change in the chemical properties of the soil with the PE MP treatments on day 28. Among the exchangeable cations, only Mg was significantly decreased by the MP_14 treatment compared to MP_0 (the control). The available N was significantly increased by the MP_7 and MP_14 treatments; however, it decreased with the MP_1 treatment. The available P was also significantly increased by the MP_7 and MP_14 treatments. The EC value was decreased by the MP_14 treatment, and the C/N ratio and pH were significantly increased by all the MP treatments compared with the control.

Table 2 shows the change in the chemical properties of the soil with the PE MP treatments on day 70. The exchangeable cations K, Na, and Mg as well as the CEC were significantly decreased by the MP_14 treatment compared to MP_0, and Ca decreased nonsignificantly. The total N significantly decreased with the MP_14 treatment, but the available N changed nonsignificantly with all of the MP treatments. The available P was significantly increased by the MP_14 treatment. The C/N ratio and pH were significantly increased by the MP_7 and MP_14 treatments compared to the initial values and MP_0. The EC values were significantly decreased by the MP_7 and MP_14 treatments.

PCA plots based on the Pearson correlation matrix were used to analyze the soil’s chemical properties after its treatment with MPs (Figure 1). The results show that the same dosages of MPs were grouped together if their soil properties were considered. In Figure 1A, F1 and F2 account for 58.08% and 14.43% of the variance, respectively. The MP_7 and MP_14 treatments were separated from the MP_0 and MP_1 treatments. The soil properties CEC, Ca, Mg, K, TN, and EC were highly correlated to MP_0 and MP_1, while the soil properties C/N, AP, pH, and AN were highly correlated to the MP_7 and MP_14 treatments.

In Figure 1B, F1 and F2 account for 60.95% and 17.09% of the variance, respectively. The soil properties Ca, Mg, TN, CEC, and EC were highly correlated to the MP_0 and MP_1 treatments, and the soil properties AP, C/N, and pH were highly correlated to the MP_7 and MP_14 treatments.

The above indicates that low concentrations of PE MPs were grouped together due to the similarities in the exchangeable cations present in soils, and high concentrations of the treatments were grouped together due to the similarities in C/N, pH, and availability of the phosphate ion.

The pH and EC responded differently in the soil and in the water after the PE MP treatments (Appendix A). It was indicated that the MP_1, MP_7, and MP_14 treatments significantly increased the pH and decreased the EC with all the MP treatments of the soils. Nevertheless, larger increases in pH and increases in EC with elevated concentrations of MPs in the water environment were recorded. The pH and EC could still be changed by the MP treatments even though the plastics had been washed before incubation, indicating that the influence of additives should be taken into consideration after water treatments.

### 3.2. Influence of PE MPs on Enzyme Activities

Figure 2 shows the soil enzyme activities during 70 days of incubation. The FDA activity showed a decreasing trend at days 1, 7, 42, and 70 but not at days 14 and 28 as the PE MP concentration increased. Low MP concentrations (1% and 7%) did not result in significant decreases in the average FDA, whereas the MP_14 treatment resulted in a significant decrease.

The p-nitrophenol release for the ACP activity during incubation showed an increasing trend with an increase in the MP concentration except on days 1 and 14. On day 1, the activity with the MP_14 treatment was significantly lower than that with the other treatments, and on days 7 and 42, the activity with the MP_14 treatment was significantly higher than that with the other treatments. The average ACP activity was significantly higher for the MP_14 treatment than that for the MP_0, MP_1, and MP_7 treatments.

The p-nitrophenol release for the NAG activity showed an increasing trend for most of the days. On days 14, 42, and 70, the activity with the MP_14 treatment was significantly increased compared to that with the MP_0 treatment. The average NAG activity was significantly higher for the MP_14 and MP_7 treatments than that for the MP_0 treatment. Overall, NAG is a more sensitive indicator compared with FDA and ACP for PE MP contamination by dose.

### 3.3. Influence of PE MPs on Microbial Community Physiological Profiles

Different carbon source utilization rates were used to determine the AWCD at 48 h for each treatment using Biolog^®^ Ecoplates^TM^ (Figure 3). On day 1, the AWCD with the MP_7 and MP_14 treatments were significantly decreased compared to that with the MP_0 treatment. On day 7, the AWCD values with the MP_1 and MP_7 treatments were significantly increased compared to that with the MP_0 treatment, whereas that with the MP_14 treatment was not significantly different from the control. On day 14, the AWCD value with the MP_14 treatment was significantly decreased compared to that with the MP_0 treatment; however, those for the MP_1 and MP_7 treatments were not significantly different. The results indicated that some microbes were inhibited by the MP treatments immediately, but some microbes that can utilize or tolerate MPs were inhibited. Over time, all the MP treatments did not cause a significant change in the microbial activity, but on average, the MP_14 treatment did result in a significant decrease during 70 days of incubation (Figure 3B).

Appendix A shows the diversity index of the substrate utilization as determined using Biolog^®^ Ecoplates^TM^ immediately after the PE MP treatments. The decrease in microbial diversity was obtained using Shannon’s diversity index, but only the MP_7 treatment showed a significant difference. The substrate richness and evenness were not significantly affected by the treatments compared to the control. However, the species richness decreased with the MP_14 treatment but increased with the MP_1 and MP_7 treatments, suggesting that some species may be inhibited under high PE MP concentrations.

Overall, we observed that the MP_1 and MP_7 treatments may promote microbial metabolism in the short term and may contribute to the growth of some specific microorganisms. Further, the MP_14 treatment reduced the soil microbial diversity immediately after its application as well as the average microbial metabolism after 70 days.

### 3.4. Influence of PE MPs on Culturable Bacteria

To evaluate the impact of PE MPs on culturable bacteria with different nutritional requirement regimes, we applied both nutrient-rich and nutrient-poor culture media in this study. Appendix A shows that the culturable bacteria increased with the MP_1 treatment but decreased with the MP_7 and MP_14 treatments in the nutrient-rich culture medium. The same trends were observed with significant differences in the nutrient-poor medium (Appendix A). The bacteria population was significantly increased by the MP_1 treatment and significantly decreased by higher PE MP concentrations. These results suggest that some culturable bacteria can use or tolerate low concentrations of PE MPs but can be suppressed by high concentrations of PE MPs. This trend was more obvious under nutrient-poor conditions. Nutrient-poor conditions may be more suitable for a cultivation-specific phylum or species of bacteria such as Alphaproteobacteria [27] or some *Pseudomonas* species [28].

### 3.5. Influence of PE MPs on Culture-Independent Bacteria

In this study, NGS was used to analyze the change in bacterial ecology after the PE MP treatments. The results show that the MP_1, MP_7, and MP_14 treatments reduced the OTU count nonsignificantly on both days (Appendix A). In Appendix A, with the MP_14 treatment, a significant decrease is observed on day 7 and an increase is observed on day 70 from the diversity index. Further, with the MP_14 treatment, a significant decrease in evenness is observed on day 7 but not on day 70. The richness did not show significant differences among the treatments. The results show that only high PE MP concentrations have a significant effect on the diversity of culture-independent bacteria. In addition, some bacteria were suppressed by the PE MP treatments, whereas bacteria that can utilize or tolerate PE MPs were augmented by the treatments after a period of time.

Figure 4 shows the relative abundance of bacteria phyla under the different PE MP treatments on days 7 and 70. Among all phyla with a relative abundance exceeding 1.0%, Proteobacteria, Acidobacteria, Actinobacteria, and Chloroflexi had the largest populations on both days. Actinobacteria was significantly increased by all the PE MP treatments on day 7. Acidobacteria was significantly decreased by the MP_7 and MP_14 treatments on day 7 and nonsignificantly decreased on day 70. Chloroflexi was significantly decreased by the MP_14 treatment on day 70.

Among the phyla with lower relative abundances, Patescibacteria were augmented by the PE MP treatment on both days 7 and 70. The relative abundance of Nitrospirae, Bacteroidetes, and Gemmatimonadetes was significantly decreased only by the MP_14 treatment on day 7. The relative abundance of Verrucomicrobia was significantly increased by all the PE MP treatments on day 70. The relative abundance of Bacteroidetes was only significantly increased by the MP_1 and MP_7 treatments on day 70.

Overall, the results showed that only the addition of 1% PE microplastics shifted the dominant phyla in the soil environment over the 70-day incubation period.

Figure 5 shows the families whose relative abundance exceeded 1.5% on days 7 and 70. Most of the families belonged to Proteobacteria and Actinobacteria. On day 7 (Figure 5A), Burkholderiaceae and Nocardioidaceae had the highest relative abundance, and on day 70 (Figure 5B), Pedosphaeraceae, Pseudomonadaceae, Nitrosomonadaceae, Burkholderiaceae, and Gemmatimonadaceae had the highest abundance. Again, only the MP_1 treatment shifted the dominant families in the soil environment over the 70-day incubation period.

On day 7, the relative abundances of Xanthomonadaceae, Solimonadaceae, Mycobacteriaceae, Streptomycetaceae, and Nocardiodaceae were significantly increased by the PE MP treatments, especially the MP_7 and MP_14 treatments. The relative abundances of Rhadocyclaceae, Pseudomonadaceae, Nitrosomonadaceae, Pyrinomonadaceae, and Gemmatimonadaceae were significantly reduced by the PE MP treatments, especially the MP_14 treatment.

On day 70, the relative abundances of Pedosphaeraceae, Solimonadaceae, and Mycobacteriaceae were significantly increased by the PE MP treatments. The relative abundances of Xanthomonadaceae, Haliangiaceae, Pyrinomonadaceae, and Gaiellaceae were significantly reduced by the PE MP treatments, especially the MP_14 treatment. Only low PE MP concentrations could induce the growth of Chitinophagaceae.

Appendix A show the PCA results of the PE MPs’ dose effects. On day 7, only the MP_14 treatment had a distinct cluster from the other treatments, whereas on day 70, the MP_1, MP_7, and MP_14 treatments all had clusters that were distinct from each other. The impact of the PE MPs on the soil bacterial ecology was such that it did not recover to the initial state over the 70-day incubation period.

Appendix A shows that basic metabolism pathways, including amino acid metabolism, carbohydrate metabolism, and energy metabolism, dominate the survival of microorganisms. Higher abundances were observed for membrane transport and nucleotide metabolism. The PE MP treatments were seen to change these metabolism pathways. For example, membrane transport was induced by the MP_14 treatment on day 70, and nucleotide metabolism was induced by the MP_1 and MP_7 treatments on day 70.

## 4. Discussion

### 4.1. Dose-Related Effect of PE MPs on Chemical Properties of Soil

PE MPs contain additives such as antioxidants, antiblocking agents, anti-adhesive agents, lubricants, and stabilizers to increase their quality, strength, and durability [29]. Additives are unavoidably released from plastics because they are not covalently bound to polymers [15,30]. In this study, the PE MPs were produced by grinding and then washed with disinfected water to imitate the possible negative effects of commercial products after weathering and leaching in the soil environment. Therefore, our results should only be compared to those of previous studies that also investigated commercial products with MPs, unless otherwise specified.

Capolupo et al. [31] found that MPs (PP, polystyrene, polyethylene terephthalate, and polyvinyl chloride) in freshwater and salt water changed the pH differently, owing to differences in leachate release. Similar concentrations of organic additives (e.g., benzothiazole, phthalide, and acetophenone) were found in the freshwater and marine leachates; however, the metal concentrations were significantly different. For example, the aluminum concentration was significantly higher in the freshwater leachate than in the marine one. However, the reasons for this difference have not been clarified. Since different soils have different properties which are more complicated than the salinity change in waters, the diverse interactions of soil environments with MPs require further investigation.

Many studies have added ground plastics into soil and determined the resulting changes in the chemical properties of the soil, especially the pH levels. These studies explained that the higher increase in the soil pH may be due to changes in the physical properties of the soil, such as its aeration and porosity [10,32,33]. Appendix A show the pH increase with both the washed and unwashed PE MP treatments, indicating that the elevated pH level might also be due to the surface properties of the PE particles as well as the release of additives. The surface properties of marine eroded PE have already been studied [34]. Its affinity to microbes was shown to be due to the increased surface area, and its polarity was shown to be due to the presence of ketone and ester groups as a result of erosion. In addition, when the surrounding pH is higher than the point of zero charge of the eroded PE (which was 6.1 in their study), the surface charge of the PE became negative. The results explained how the surface properties of ground PE affected soil properties such as the pH level and availability of nutrient elements. 

The soil showed a decrease in cations, such as K, Na, Ca, and Mg, and an increase in anions, such as nitrate and phosphate ions. The additives in plastics may contain P, N, and Cl, which impact the chemical properties of the soil. However, the change in ions in the soil may be explained by the surface charge of the ground PE as well [29]. Based on the PCA (Figure 1 and Figure 2), PE MP treatments with low concentrations were different from other treatments in terms of exchangeable Ca and Mg, which probably is due to the negative surface charges on the microplastics. However, at high concentrations, the pH and the availability of the phosphate ion could be due to the additives. Furthermore, the soil properties showed a dose-related effect from the MP treatments. The changes in the soil remain over the 70-day incubation period with only the MP_1 treatment, indicating that MPs may have long-term impacts.

Nevertheless, some studies have reported conflicting findings. For example, in one study, as the concentration of plastic residues increased in a cotton field, the soil pH increased and available K and P decreased [35]. Qi et al. [36] noted that 2% LDPE MPs did not have any significant effects on the soil pH and EC during a 30-day incubation period. Another study reported that a 1% LDPE treatment did not result in any significant differences in pH [37]. Because the soil properties can also be influenced by other factors, such as the moisture content, organic matter content, and other physiochemical properties [10,12,38,39,40], further studies must be conducted to predict the possible impact of soil property changes from PE MP treatments both in the laboratory and in the field.

### 4.2. Dose-Related Effect of PE MPs on Extracellular Enzymatic Activity in Soil

The soil enzyme activity plays a key role in nutrient turnover and can provide information about the impact of exogenous disturbances in soils [41]. In this study, we measured the effect of PE MP treatments on the activity of three soil enzymes. The FDA activity represents the overall microbial metabolic activity from decomposers, bacteria, and fungi [42]. The ACP enzyme is involved in microbial cycling and the transformation of organic P into inorganic forms in soil [43]. The NAG enzyme is involved in chitin degradation, which plays a key role in C and N cycling [44,45]; the effects of PE MP treatments on the NAG enzyme have not been reported yet.

MPs can provide a habitat for microbes [34]; however, the average FDA activity decreased with the MP_14 treatment in this study (Figure 2). Further, the average AWCD also decreased with the MP_14 treatment (Figure 2), and the decrease exhibits a clear correspondence with the increase in PE MP concentration. Low PE MP concentrations (1% and 7%) did not have a significant effect on the microbial activity in the long run. Unsurprisingly, high PE MP concentrations from commercial resins leached higher amounts of additives, and thus had a larger impact on the chemical properties of the soil (Appendix A). However, previous studies indicated that 28% pure-grade PE MPs also inhibited enzyme activity by changing the nutritional substrates and physicochemical properties of the soil through adsorption [39], while FDA was inhibited by a 5% pure PE MP treatment [38]. High PE MP concentrations in both the pure form and with additives in the soil environment had negative effects on the soil’s microbe metabolism. The changes in the physical and chemical properties of the soil could explain this decline with the addition of the additives.

The ACP activity during incubation had an increasing trend with an increase in the MP concentration, except on days 1 and 14 (Figure 2). The P-containing additives may explain the increase in ACP activity. However, 5% pure-grade PE MPs were reported to increase ACP activity [38] because of the higher moisture content in PE-MP-treated soils. PE MPs in the pure form or with additives increased the soil ACP activity in the long run with a dose-related trend.

The NAG activity was stimulated by the MP_7 and MP_14 treatments on specific days and overall (Figure 2). The effect of PE MP treatments on NAG has not yet been reported. However, other glycosidases such as ß-glucosidase have been investigated, and their effects are opposite to those found in this study [39]. Rillig et al. [46] noted that a broad range of additives from MPs can have diverse effects, such as changing the soil’s electrochemistry properties and increasing the molecular diversity of available organic C and N for microbial activity. These factors may contribute to the increased NAG activity observed in this study. Increased relative abundances of the Nocardioidaceae family, which degrades chitin, starch, and casein and uses N-acetylglucosamine as a sole carbon source, was also seen in this study (Figure 5) [47]. The results showed that the addition of PE MPs enhanced the NAG enzymatic activity due to the shift in related functional microbes, and this may consequently contribute to the change in C and N cycling in the soil. The average NAG seemed to correspond more sensitively to the change in PE MP concentration.

The FDA, ACP, and NAG activities exhibited a dose-related trend with the addition of PE MPs. However, only the MP_14 treatment significantly affected the average FDA and ACP activities, and the MP_7 and MP_14 treatments significantly increased the average NAG activity.

### 4.3. Culturable Bacteria Can Be Augmented by MP Treatments

The interactions between PE MPs and the soil environment showed that different treatments can differently impact the microbial communities in soil (Figure 2A and Figure 3). Culturable bacteria were observed using both Biolog^®^ Ecoplates^TM^ and a culture medium. The increase in metabolic potential can be related to bacterial communities that are able to use PE MPs and their additives as carbon sources [48] or that can adapt to changes in soil properties such as aeration, porosity, pH, and element levels [10,32,33].

Both experiments show that PE MPs have a dose-related impact on culturable bacteria. A low dosage of PE MPs can increase the bacterial population, whereas a high dosage had a detrimental effect. We assumed that a high dosage had more pronounced effects on the release of additives, whereas at lower dosages, the surface electrostatic forces can attract some culturable bacteria that can adapt to or even utilize the PE MPs and their additives. A comparison of Appendix A reveals more pronounced changes in oligotrophic bacteria. Nutrient-poor conditions may be more suitable in cultivating specific bacteria such as Alphaproteobacteria [27] or some *Pseudomonas* species [28]. These bacteria were also abundant in the culture-independent experiment in this study. Pseudomonadaceae were augmented nonsignificantly by the MP_7 treatment on day 70 (Figure 5B).

### 4.4. Influence of PE MPs on Culture-Independent Bacteria Phyla

The biodiversity of culture-independent bacteria can be revealed by applying NGS techniques [49]. For culture-independent bacteria, low PE MP concentrations did not have a significant effect on the soil microbial diversity in the long term (Appendix A), as also reported by Huang et al. [48] but contradicted by Fei et al. [38].

However, the changes in microbial communities were evident at the phylum level even with low PE MP concentrations (Figure 4). The phyla with the highest relative abundance on both days of the incubation period included Proteobacteria, Actinobacteria, Acidobacteria, and Chloroflexi. These phyla are common in soil environments [38,50].

The Proteobacteria phylum, which is the most common in the domain of bacteria [51], decreased nonsignificantly with the MP_1 and MP_14 treatments on day 7 and with all of the MP treatments on day 70 of the incubation period (Figure 4). The decreases in the relative abundances of the Proteobacteria phylum are concerning because some members can degrade PE, therefore limiting the biodegradation of PE MPs in the soil [52].

Acidobacteria, a phylum that is also common in all terrestrial ecosystems, can participate in many vital ecological processes and promote the growth of various plants. However, many of them can only grow in oligotrophic environments [53]. In this study, we found that the abundance of Acidobacteria was significantly decreased by the addition of PE MPs in a dose-related manner; this finding agrees with that of one study that applied PE to alkaline soil (with a pH of 7.62) [54] and another study that applied PE to acidic soil (with a pH of 5.5) [38].

Most Actinobacteria are widely distributed in both terrestrial and aquatic ecosystems and are free-living. They produce diverse metabolites with various bioactivities [55]. The Actinobacteria phylum has been reported to be predominant after the addition of PE MPs; in contrast, the Proteobacteria phylum was predominant in the control [54]. Further, in our study, the Actinobacteria phylum was enriched by the addition of PE MPs in a dose-related manner, possibly owing to the biodegradation of PE MPs by some species of Actinobacteria [50,56,57].

The Chloroflexi phylum, an anoxygenic phototroph, has primarily been associated with extreme habitats and hypersaline environments [58]. The decline in its population after a PE MP treatment was reported to be due to the reduction in N_2_O emissions [54]. In this study, inhibition was observed on day 70.

The superphylum Patescibacteria has been found to be prevalent in groundwater, sediment, lakes, and other aquifer environments. These bacteria have ultra-small cells with reduced redundant and nonessential functions, and they can survive in environments with low concentrations of nutrients (including C, N, S, and P) [59]. They have been reported to play an important role in breaking down the vinyl ester-base polymer, and they can be enriched on the surface of PE particles [60]. A population was observed to be augmented by PE MPs in a dose-related manner, similar to Actinobacteria.

Bacteroidetes can colonize all habitats, including soils, oceans, and freshwater. They are one of the main members of the microbiota in the animal gastrointestinal tract, and they play a role in the degradation of organic matter [61]. Bacteroidetes were decreased by the MP_14 treatment on day 7. Fierer et al. [62] noted a decrease in Acidobacteria and an increase in β-Proteobacteria and Bacteroidetes after the addition of carbon. The decrease in Bacteroidetes after the addition of PE MPs may suggest the loss of ecosystem functions that were once provided by the phylum.

### 4.5. Influence of PE MPs on Culture-Independent Bacteria Family

In this study, a shift in the bacterial ecology in a culture-independent way was observed on days 7 and 70 in a dose-related manner (Figure 5), suggesting that the impact can be augmented if MP pollution worsens. MPs have been reported to impact soil microbes directly or indirectly through changes in soil properties, and their presence may also impact nutrient cycling [13]. These changes can be further confirmed by looking at the differences in basic metabolism pathways (Appendix A).

The bacterial families Xanthomonadaceae, Rhadocyclaceae, Pseudomonadaceae, Burkholderiaceae, Nitrosomonadaceae, and Pseudomonadaceae from the Proteobacteria phylum were affected differently by the PE MPs on days 7 and 70 (Figure 5). Many species in these families have been described to have nitrogen fixation capabilities and to be involved in C and N cycles [56,63,64,65,66]. The relative abundances of the Rhadocyclaceae family (e.g., genera *Azoarcus*, *Azovibrio*, and *Thauera*), Pseudomonadaceae family (e.g., genera *Azotobacter* and *Pseudomonas*), and Nitrosomonadaceae decreased with the MP_14 treatment on day 7 but was nonsignificant on day 70. The Burkholderiaceae family showed slight increases in the relative abundance on day 7 with the MP_7 and MP_14 treatments; however, the relative abundances with all the MP treatments were slightly lower than that of the control on day 70. The augmentation in the Burkholderiaceae family immediately after the addition of PE MPs was consistent with other studies that investigated 7% pure PE MPs after 50 days [38] or commercial PE MP powder after 42 days [67]. However, in this study, the high PE MP concentration inhibited the Burkholderiaceae family on day 70. Most of the above-mentioned families that played a role in N cycling were decreased by their exposure to PE MPs, suggesting a further need to investigate their impact on N cycling.

Bacterial families from the Actinobacteria phylum were also affected differently by the PE MP treatments on days 7 and 70 (Figure 5). The Mycobacteriaceae, Streptomycetaceae, and Nocardioidaceae families in the Actinobacteria phylum were positively correlated with the MP_14 treatment on day 7 (Appendix A). However, only Mycobacteriaceae was augmented on day 70. Some members in this family were associated with the degradation of carbon-based pollutants [68] and the synthesis of urea [69].

Pedosphaeraceae was significantly induced by all of the PE MP treatments on day 70. It was reported to play an important role in sediment nutrient circulation and to influence metal resistance potentials [70]. It can thrive in environments that have limited resources and plays an important role in soil organic carbon accumulation and N stock [71,72]. The relative abundance of the Haliangiaceae family decreased with both the MP_7 and MP_14 treatments on day 70; consequently, the vital role this family plays in the turnover of carbon in soil ecosystems was affected [73,74]. The relative abundance of the Solimonadaceae family was largely augmented on days 7 and 70. This family thrives under low-nutrient conditions and makes P bioavailable in soil [75,76], thus influencing the P cycle. Two bacteria families in the Acidobacteria phylum had high relative abundances (>1.5%) after the addition of PE MPs. Pyrinomonadaceae, a major family in the Acidobacteria phylum [77], decreased over the 70-day incubation period. The population of the Blastocatellaceae families also decreased on day 7 but recovered by day 70. Some members of the Blastocatellaceae family were reported to be able to degrade complex carbon compounds. Overall, a shift in the functional microbes responsible for carbon utilization and cycling was observed.

A metagenomic study reported that the Sphingomonadaceae, Bryobacteraceae, Chitinophagaceae, Caulobacteraceae Cyclobacteriaceae, and Gemmatimonadaceae families contained many phosphate-solubilizing bacteria [78]. Among them, the Sphingomonadaceae, Chitinophagaceae, and Gemmatimonadaceae families had relative abundances exceeding 1.5% in this study. Sphingomonadaceae showed no significant change with the PE MP treatments. Gemmatimonadaceae showed a significant decrease only on day 7. The addition of PE MPs induced the growth of Chitinophagaceae only at low concentrations. It was reported that Chitinophagaceae, which can be related to phosphate solubilization [78], were inhabited on PE MPs in both marine and freshwater environments [79]. Overall, a shift in the functional microbes responsible for phosphate utilization and cycling was observed.

PE MPs in soil influence the microbial community structure depending on their concentration and the changes induced in the soil’s properties by their addition. Some families that can usually survive in polluted environments and with limited resources were affected; this finding was similar to the results derived from culturable bacteria. A higher PE MP concentration had a greater impact on the soil properties and microbial ecology, and this impact was constant throughout the experiment. These observations, together with the fact that soil is a sink for long-lasting MPs, highlight the importance of remediating MP pollution and the need for further knowledge on related subjects.

## 5. Conclusions

This study revealed that PE MPs changed the soils’ properties, enzymatic activities, and bacterial community diversity, structure, and functions in a dose-related manner. High concentrations of 7% and 14% MPs resulted in changes to the C/N ratio, pH, and available phosphate ions. Nevertheless, only the MP_14 treatment significantly influenced the average FDA and ACP activities, whereas both the MP_7 and MP_14 treatments significantly enhanced the average NAG activity. Some culturable bacteria can utilize or tolerate low PE MP concentrations while being suppressed by high PE MP concentrations. This trend was more obvious under nutrient-poor conditions. Only a high PE MP concentration may have a significant impact on the diversity of culture-independent bacteria; however, analyses at the phylum and family levels revealed shifts in the soil microbial ecology. Various genera of bacteria involved in biogeochemical cycles were radically changed, suggesting that PE MPs in soil may alter the C, N, and P cycles. Such changes were observed throughout the experiment, suggesting that PE MPs may have a long-term impact on the soil environment with a dose-related magnifying effect.

## Figures and Tables

**Figure 1 microorganisms-12-01790-f001:**
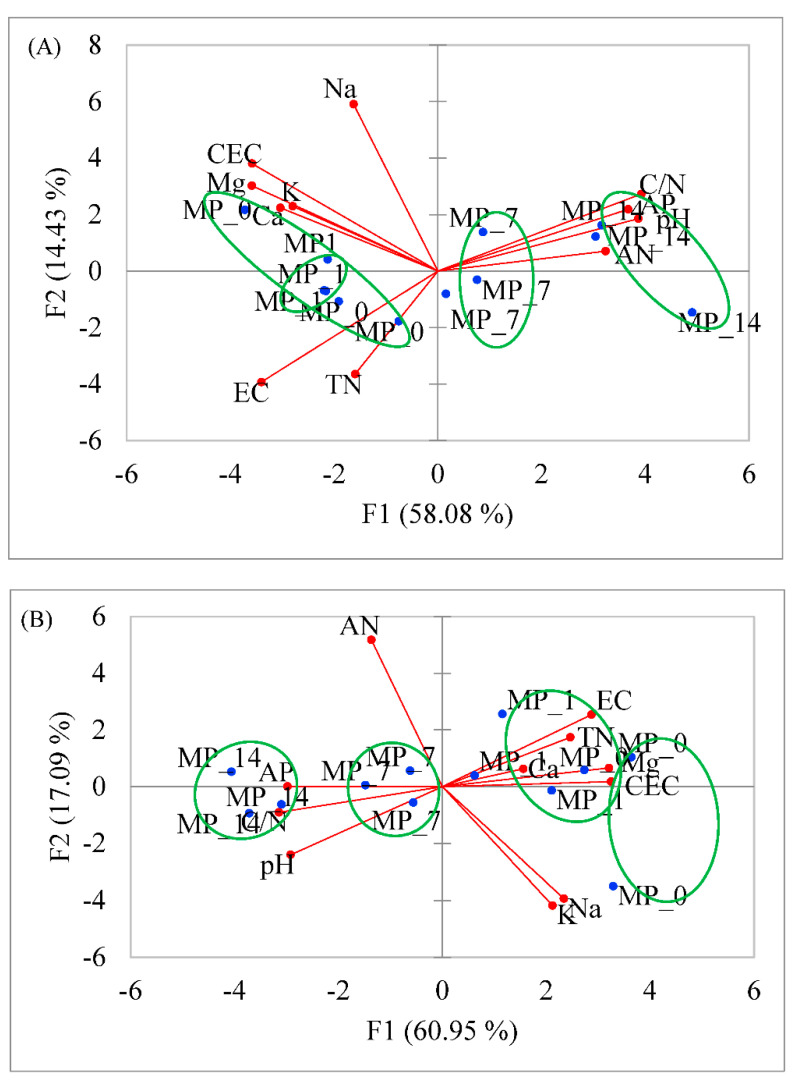
PCA of soil chemical properties at (**A**) day 28 and (**B**) day 70 after PE MP treatments.

**Figure 2 microorganisms-12-01790-f002:**
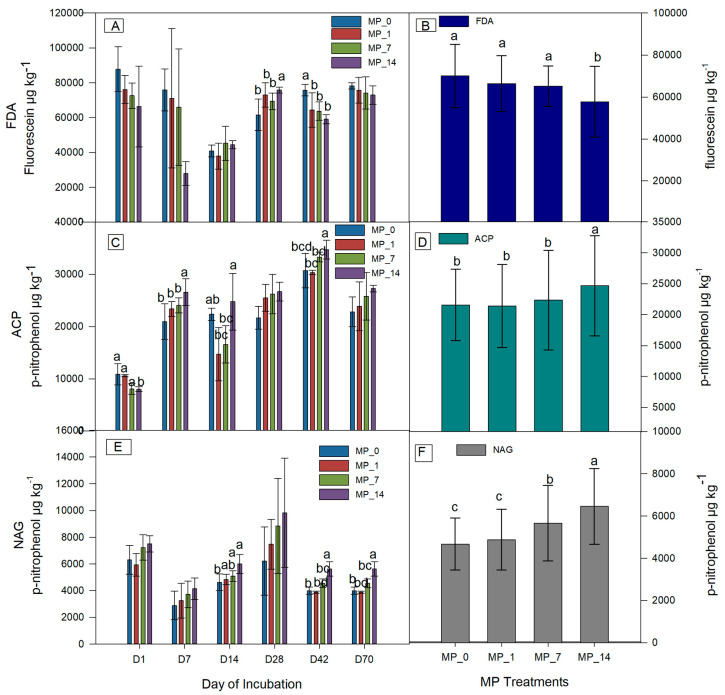
Soil enzyme activities of (**A**) FDA; (**B**) average FDA; (**C**) ACP; (**D**) average ACP; (**E**) NAG; (**F**) average NAG at days 1, 7, 14, 28, 42, and 70 under control (MP_0), MP_1, MP_7, and MP_14 treatments. Different letters denote significant differences among treatments (*p* < 0.05).

**Figure 3 microorganisms-12-01790-f003:**
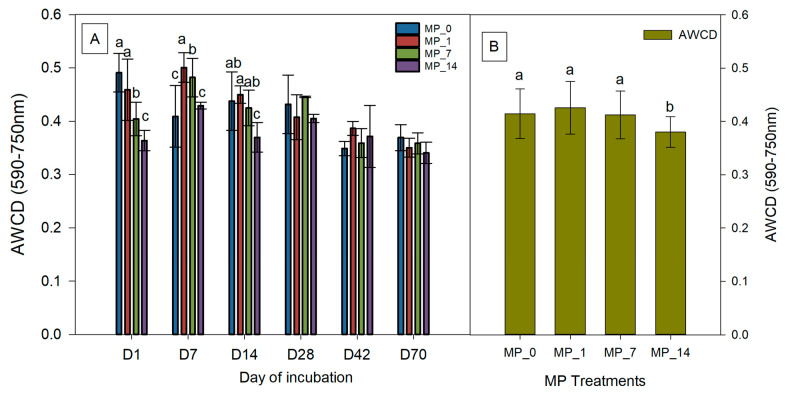
(**A**) AWCD and (**B**) average AWCD values at days 1, 7, 14, 28, 42, and 70 under control (MP_0), MP_1, MP_7, and MP_14 treatments. Different letters denote significant differences among treatments (*p* < 0.05).

**Figure 4 microorganisms-12-01790-f004:**
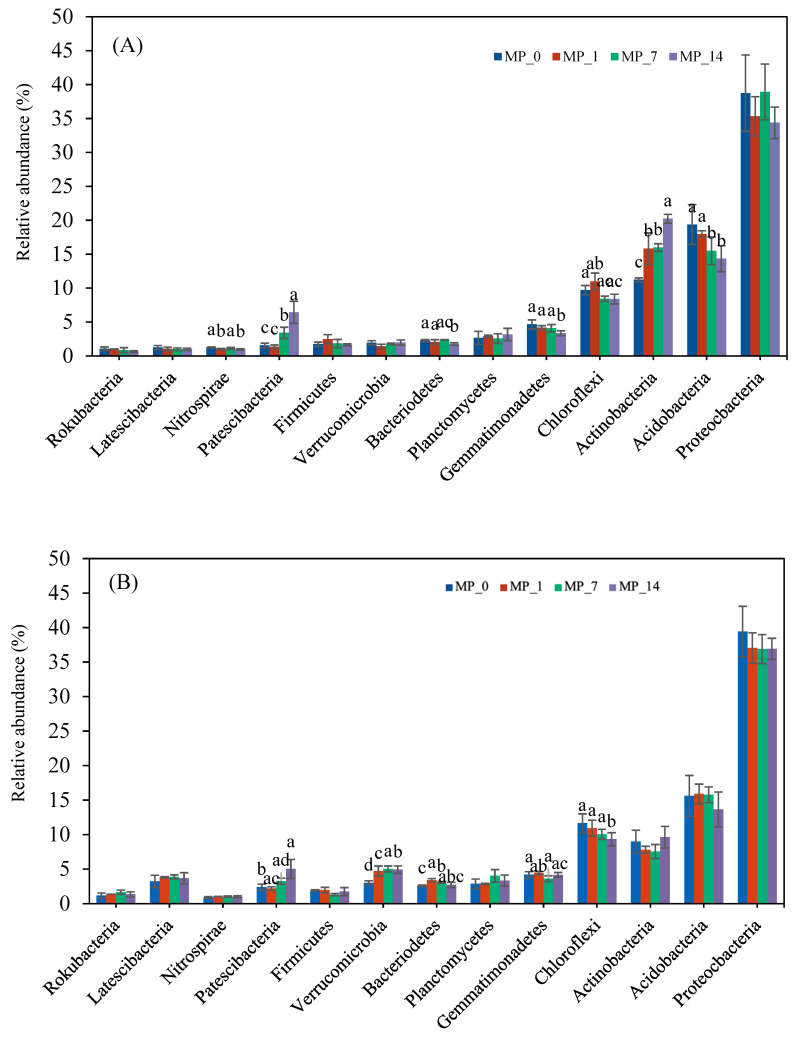
Phylum changes with PE MP treatments at (**A**) day 7 and (**B**) day 70. Different letters denote significant differences among treatments (*p* < 0.05).

**Figure 5 microorganisms-12-01790-f005:**
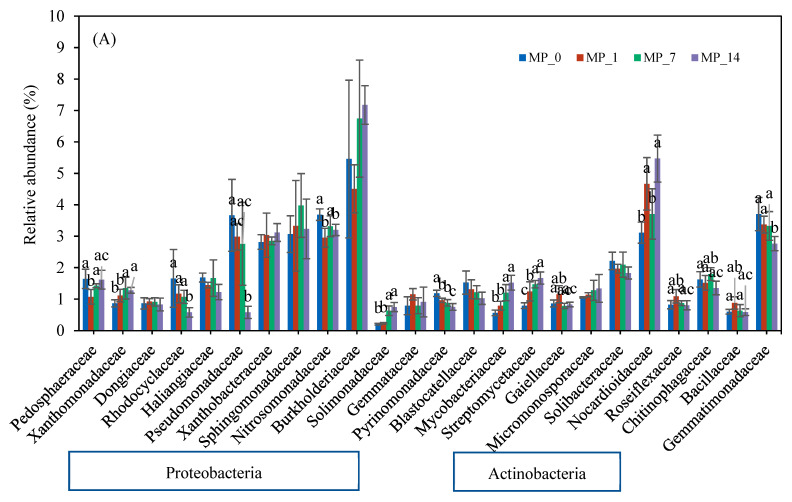
Family changes with PE MP treatments at (**A**) day 7 and (**B**) day 70. Different letters denote significant differences among treatments (*p* < 0.05).

**Table 1 microorganisms-12-01790-t001:** Changes in soil chemical properties at day 28.

	Soil Properties †										
	K	Na	Mg	Ca	Total N	C/N	N	P	CEC	pH	EC
	mg kg^−1^	mg kg^−1^	mg kg^−1^	mg kg^−1^	%		mg kg^−1^	mg kg^−1^	cmol (+) kg^−1^		µS cm^−1^
Initial Value	12.0 ± 0.7 a ‡	649.3 ± 16.6 a	17.1 ± 1.4 a	114.0 ± 25.4 a	0.04 ± 0.01 b	24.7 ± 2.7 cd	1.9 ± 0.2 d	1.73 ± 0.15 a	285.1 ± 15.8 a	6.73 ± 0.02 d	156.2 ± 3.4 b
MP_0	11.6 ± 1.7 ab	609.2 ± 61.9 ab	15.7 ± 1.7 a	80.1 ± 6.4 b	0.05 ± 0.01 ab	19.9 ± 3.5 d	4.0 ± 0.7 b	0.96 ± 0.05 c	206.5 ± 20.5 bc	6.84 ± 0.06 d	235.3 ± 16.3 a
MP_1	11.2 ± 1.6 ab	594.0 ± 15.8 ab	16.9 ± 0.6 a	78.9 ± 3.2 b	0.06 ± 0.01 a	31.8 ± 6.1 c	3.1 ± 0.4 c	1.09 ± 0.02 c	214.5 ± 3.0 c	6.92 ± 0.02 c	236.3 ± 13.7 a
MP_7	9.9 ± 1.0 ab	590.0 ± 27.8 ab	15.3 ± 0.5 ab	74.0 ± 2.8 b	0.05 ± 0.01 ab	127.7 ± 43.0 b	4.5 ± 0.4 a	1.16 ± 0.09 b	200.3 ± 3.8 bc	7.01 ± 0.07 b	214.7 ± 16.5 a
MP_14	9.7 ± 0.6 b	588.8 ± 11.8 b	12.9 ± 2.5 b	66.9 ± 11.7 b	0.05 ± 0.00 ab	208.8 ± 6.2 a	5.2 ± 0.4 a	1.31 ± 0.05 b	179.5 ± 23.6 b	7.19 ± 0.04 a	156.1 ± 17.0 b

Note: † K, Na, Mg, and Ca were determined in the exchangeable form. N and P were determined in the available form. CEC: Cation exchange capacity, EC: Electrical conductivity. ‡ Letters a–d following the mean ± SD values denote difference of mean at significant level of 0.05 by LSD post-hoc test.

**Table 2 microorganisms-12-01790-t002:** Changes in soil chemical properties at day 70.

	Soil Properties †										
	K	Na	Mg	Ca	Total N	C/N	N	P	CEC	pH	EC
	mg kg^−1^	mg kg^−1^	mg kg^−1^	mg kg^−1^	%		mg kg^−1^	mg kg^−1^	cmol (+) kg^−1^		µS cm^−1^
Initial Value	12.0 ± 0.7 a ‡	649.3 ± 16.6 a	17.1 ± 1.4 a	114.0 ± 25.4 a	0.04 ± 0.01 b	24.7 ± 2.7 c	1.9 ± 0.2 a	1.73 ± 0.15 a	285.1 ± 15.8 a	6.73 ± 0.02 c	156.2 ± 3.4 d
MP_0	10.0 ± 3.2 a	104.7 ± 13.4 b	15.3 ± 0.5 a	78.8 ± 15.9 b	0.07 ± 0.01 a	14.8 ± 0.8 c	1.8 ± 0.4 a	1.05 ± 0.07 c	148.1 ± 3.4 b	6.73 ± 0.06 c	288.7 ± 32.0 a
MP_1	8.3 ± 1.0 ab	89.9 ± 13.7 b	14.1 ± 0.5 b	68.3 ± 3.9 b	0.06 ± 0.02 a	30.7 ± 7.7 c	2.3 ± 0.4 a	1.12 ± 0.08 c	135.2 ± 5.0 b	6.71 ± 0.02 c	299.3 ± 6.7 a
MP_7	7.2 ± 0.7 ab	84.7 ± 5.8 bc	12.8 ± 0.4 bc	71.1 ± 8.1 b	0.06 ± 0.02 ab	114.2 ± 25.6 b	2.0 ± 0.0 a	1.18 ± 0.13 c	123.9 ± 3.8 bc	6.83 ± 0.04 b	239.0 ± 14.1 b
MP_14	6.3 ± 0.2 b	82.1 ± 2.3 c	12.0 ± 0.4 c	67.7 ± 6.5 b	0.04 ± 0.01 b	285.2 ± 45.7 a	2.3 ± 0.4 a	1.31 ± 0.05 b	116.6 ± 3.0 c	6.98 ± 0.03 a	205.6 ± 13.3 c

Note: Footnotes resemble those in Table 1.

## Data Availability

The raw data supporting the conclusions of this article will be made available by the authors on request.

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
