# Peer review of "Dose Effect of Polyethylene Microplastics Derived from Commercial Resins on Soil Properties, Bacterial Communities, and Enzymatic Activity"

_microorganisms, 2024, doi:10.3390/microorganisms12091790_

Round 1
Reviewer 1 Report
Comments and Suggestions for Authors
The Manuscript ID: ID: microorganisms-3157146 examines dose effect of polyethylene microplastics derived from commercial resins on soil properties, bacterial communities, and enzymatic activity changes. As polyethylene microplastics (PE MPs) contain additives such as antioxidants, antiblocking agents, anti-adhesive agents, lubricants, and stabilizers, they are supposed to have direct/indirect impacts on the tested targets. Interestingly, the authors produced PE MPs by grinding and then washed with disinfected water to imitate the possible effects suffered by commercial products after weathering and leaching in the soil environment. For example, among soil properties, the pH increase with both washed and unwashed PE MPs treatments might be due to the surface properties of the PE particles and the release of additives. They found that PE MPs treatments with low concentrations could be separated from other treatments by differences such as in exchangeable Ca and Mg, whereas at high concentrations, the pH and the available phosphate ion were the major causes. Also, the fluorescein diacetate (FDA), acid phosphatase (ACP), and N-acetyl-β-d-glucosaminidase (NAG) enzyme activities showed a dose-related trend with PE MPs amendment. Yet, the average FDA and ACP activities could be significantly affected only by MP_14. Changes in the microbial communities were noted at both the phylum and family levels with all PE MPs treatments. Their data indicated that even the low dosage of PE MPs in soils can affect the functional microbes and greater impact can be noted on those that can survive in polluted environments with limited resources. Eventually, PE MPs could impact the soil properties, soil enzymatic activities, and soil bacterial community diversity, structure, and functions in a dose-related manner. The study addresses an important and interesting subject as it suggests that PE MPs may have a long-term impact on the soil environment with a dose-related boosted effect.
Generally, the subject is worth publication and the authors did a good job. However, a few typos and defects should be considered, to name but a few:
1) In the meantime, different doses of MPs ranged from industrial contamination level 68 to the farm land accumulative level were applied.
2) Differential concentrations of MPs (0%, 1%, 7%, and 14%) which had been washed with disinfected H2O or without washing were mixed
3) Figures such as Fig. 1. (title or caption should be put under it for its own explanation).
Therefore, I would suggest accepting it after minor revision.
Reviewer 2 Report
Comments and Suggestions for Authors
The reviewed work falls within the framework of the so-called ‘topical subjects’ of great interest. The work seems well thought out and consistently presented, but in its present form it cannot be published as it has some inaccuracies and needs to be corrected and supplemented. I attach some comments below:
1. titles of figures are missing throughout the work.
2. please explain the method of water disinfection? Are the authors sure that they used disinfected water or perhaps sterile water?
3. What is the difference between Initial Value and MP_0? Initial Value appears in the tables.
4. in the summary, the term hight level of MPs should be clarified, e.g. by indicating in brackets the highest concentration analysed.
5. It is worth extending this study to an analysis of the components and their amounts that can be leached from the plastic, especially as the authors analysed changes in e.g. pH, EC of the aqueous phase in contact with water. With such results, one can confidently analyse and correlate with the observed changes in the results.
Round 2
Reviewer 2 Report
Comments and Suggestions for Authors
I have no comments on the revised work.
Author Response
Thank you very much!